# Common Demand vs. Limited Supply—How to Serve the Global Fight against COVID-19 through Proper Supply of COVID-19 Vaccines

**DOI:** 10.3390/ijerph19031339

**Published:** 2022-01-25

**Authors:** Yiqing Su, Yanyan Li, Yanggui Liu

**Affiliations:** 1Regional Social Governance Innovation Research Center, Guangxi University, Nanning 530004, China; 2School of Public Policy and Management, Guangxi University, Nanning 530004, China; liyanyan@st.gxu.edu.cn (Y.L.); liuyanggui@st.gxu.edu.cn (Y.L.)

**Keywords:** global public goods, COVID-19 vaccines, institutional design, nested institution

## Abstract

Vaccination plays an essential role in the fight against Coronavirus Disease 2019 (COVID-19). The currently insufficient vaccine production capacity makes it difficult to balance supply with demand, which has led to a contradiction between command demand and limited supply. According to analysis based on game theory, the attributes of COVID-19 vaccines vary with supply strategies formulated by vaccine-producing countries. This means that vaccine-receiving countries can only be motivated to prepare operable vaccine distribution plans through the supply of COVID-19 vaccines as global public goods. The rational distribution of global public goods must be guaranteed by a global supply institution system. To that end, Elinor Ostrom’s eight design principles provide a basis for designing such a global supply system. This paper proposes a nested institution solution for guaranteeing the global supply of COVID-19 vaccines based on the design principles, which include clearly defined boundaries, proportional equivalence between benefits and costs, collective-choice arrangements, monitoring, graduated sanctions, conflict-resolution mechanisms, minimal recognition of rights to organize, and nested enterprises. To win this global fight against COVID-19, COVID-19 vaccines must not only be treated as global public goods, but countries must also be urged to coordinate cooperation in global institutional design, thus ensuring that COVID-19 vaccines can truly benefit all mankind.

## 1. Introduction

From the outbreak of the coronavirus disease 2019 (COVID-19) to 20 November 2021, more than 250 million confirmed COVID-19 cases were reported in over 200 countries and regions, resulting in a cumulative death toll of 5.15 million. The global crisis triggered by COVID-19 has led to the most serious global economic recession since World War II [1]. The COVID-19-imposed global public health emergency has spread quickly across the world within a very short period, posing persistent threats to human life and property security across both time and space [2]. A series of measures (such as case isolation and traffic barring) have been taken to interrupt virus spread routes, which have proved vital in inhibiting virus spread; however, such measures are not long-term solutions. To fundamentally control SARS-CoV-2 infection and effectively restore normal order worldwide, large-scale vaccination must be promoted for the acquisition of antibodies [3]. Consequently, universal vaccination with safe and effective vaccines offers the only scientific approach for the fundamental control of the global COVID-19 crisis and the restoration of normal order worldwide.

As of November 2021, the majority of countries across the globe still had no vaccine research and development or production capacity. Only 10 COVID-19 vaccines, developed by China, the USA, the UK, and Russia, had been approved for market launch, and all other countries had to purchase vaccines from them. So far, the global production capacity of commercially available COVID-19 vaccines is estimated to be 5.9 billion doses; however, the global demand for COVID-19 vaccines has exceeded 11 billion does. It is therefore foreseeable that, over the next two or three years, a large gap will remain in the global production capacity of COVID-19 vaccines, and COVID-19 vaccines will remain a scarce resource with limited supply for some time to come [4,5,6].

Regarding COVID-19 governance, different countries advocate different governance philosophies, and vaccine-producing countries have implemented different vaccine supply strategies. In general, these vaccine supply strategies can be classified into two categories. The supply strategies of the first category treat COVID-19 vaccines as national private goods and implement a market-oriented approach for the allocation of this scarce resource. For instance, certain countries assign the power of vaccine pricing to the market and implement a policy of preference to their own citizens. This approach leads to soaring vaccine prices, where only the highest bidders can afford the vaccines. In face of persistent demand and limited production capacity, countries all over the world begin to compete for limited vaccine resources. This situation not only induces feelings of powerlessness and frustration in developing countries, but it also triggers various kinds of conflicts, such as unilateral breaching of vaccine supply contracts by vaccine producers. The supply strategies of the second category regard COVID-19 vaccines as global public goods and give due consideration to vaccine accessibility and affordability in various countries of the world. For instance, the COVID-19 Vaccines Global Access (COVAX) initiative provides vaccines to countries that have joined the initiative according to certain distribution principles, and thus helps developing countries obtain COVID-19 vaccines. China has supplied COVID-19 vaccines to 110 countries and international organizations. This direct supply reduces the marginal costs of vaccine production and allocation, it and promotes fair distribution of vaccines worldwide.

Apparently, against the background of the global fight against COVID-19, a contradiction exists between the common demand for COVID-19 vaccines on part of most countries, and the limited supply of COVID-19 vaccines by only a few countries. The key to resolving this contradiction lies in the attitudes and strategies of vaccine-producing countries towards vaccine supply. In this case, it is worth exploring which vaccine supply strategy can better relieve the contradiction between command demand and limited supply, thus boosting the global fight against COVID-19. In view of this, this paper takes the attributes of COVID-19 vaccines as economic goods as the point of penetration, examines the different attributes manifested by COVID-19 vaccines under different strategies, and introduces game theory to explore which vaccine supply strategy can better help to win the global fight against COVID-19. On this basis, this paper further discusses how to design effective supply strategies, aiming to enrich current understanding of the global fight against COVID-19.

## 2. Attributes of Economic Goods and Supply Strategies of COVID-19 Vaccines

### 2.1. Classification and Supply Modes of Economic Goods

While many scholars have classified economic goods based on different criteria, there is a general tendency to categorize public goods as economic goods [7,8,9]. Over the past few years, a consensus has gradually been reached regarding the connotations of public goods through constant debates and exchanges among scholars from all over the world [10,11,12]. Public goods are defined as items with commonality. Specifically speaking, regardless of the way the concept of public goods is extended, mainstream research still characterizes public goods from two dimensions, namely, exclusiveness and competitiveness. Depending on their exclusiveness and competitiveness, items can be classified into public goods, common pool resources, club goods, and private goods (see Table 1). In the broad sense, public goods cover pure public goods, club goods, and common pool resources. To be specific, pure public goods are non-exclusive and non-competitive, club goods are exclusive but non-competitive, and common pool resources are competitive but non-exclusive. In addition, private goods have the attributes of both exclusiveness and competitiveness.

In general, under the market mechanism, the supply and demand of private goods spontaneously reach an equilibrium state based on the intentions of the demand and supply sides, thus realizing the effective allocation of resources. However, regarding the supply of public goods, the market constantly fails, resulting in failure to provide pure public goods and over-consumption of common pool resources. Such problems make it impossible for the market to realize effective supply of public goods based on its own resource allocation mechanism [13]. To solve these problems, scholars have developed specific governance ideas concerning specific types of economic goods (see Table 2). First, regarding the theory of pure public goods proposed by Samuelson, most scholars assume that government support is essential in the supply of pure public goods, and that pure public goods should be supplied by the government, as government interventions can compensate for market deficiencies [14,15]. Second, regarding the concept of club goods described by Buchanan, scholars tend to focus on how to transform club goods into private goods through converting the non-competitiveness of club goods into competitiveness, so that club goods can be effectively supplied by the market-oriented approach intended for supplying private goods [16,17,18]. Third, regarding common pool resources, Ostrom proposed to adopt an institutional design suitable to local conditions to urge the users of common pool resources to form effective collective actions, with the goal to realize long-term sustaining and effective governance of common pool resources [19]. Furthermore, private goods should be supplied by the market. This is because private goods have the attributes of both exclusiveness and competitiveness, and market mechanisms can spontaneously balance the supply and demand of private goods.

While the consumption of public goods is non-competitive and non-exclusive, the classification of public goods is not absolute. Changes in related external factors may modify competitiveness and exclusiveness, making it possible for economic goods to be transferred from one classification to another under different conditions [20]. Given that different economic goods correspond to different supply modes, with mutual transformation between economic goods under changing external conditions, the corresponding supply modes will change simultaneously.

### 2.2. Effects of Supply Strategies on the Attributes of COVID-19 Vaccines

Based on the two core dimensions of goods (i.e., exclusiveness and competitiveness), different supply strategies endow COVID-19 vaccines with different attributes. Currently, the extreme scarcity of COVID-19 vaccines makes them competitive on a global scale. The two existing supply modes for COVID-19 vaccines in the international community, i.e., open supply and non-open supply, will determine the exclusiveness of COVID-19 vaccines. Thus, the attributes of COVID-19 vaccines vary with different supply strategies.

#### 2.2.1. Attributes under the Non-Open Supply Strategy: Private Goods

Affected by the profit-seeking tendency of capital, the non-open supply strategy leaves COVID-19 vaccines to market-oriented distribution at the global level. The high vaccine prices thus caused have led to high exclusiveness and high competitiveness of COVID-19 vaccines worldwide. As a result, COVID-19 vaccines have gradually become national private goods of vaccine-producing countries. In this case, vaccine-demanding countries can only obtain vaccines through market purchase. However, given that vaccine supply is far lower than vaccine demand and that vaccine producers firmly seize the power of vaccine pricing, highly competitive rich countries will acquire the majority of vaccines under the market mechanism, while low-income countries will be denied access to urgently needed vaccines [21,22,23,24].

#### 2.2.2. Attributes under the Open Supply Strategy: Common Pool Resources

Common pool resources possess low exclusiveness of resource systems and high competitiveness of resource units [25]. Under the open supply strategy, all countries and individuals distributed over the world are consumers of COVID-19 vaccines; thus, low exclusiveness is manifested at the boundary of resource systems. Because of the extreme scarcity of COVID-19 vaccines in the foreseeable future, the competition for limited resource units will become very intense (Figure 1).

Specifically, from the perspective of exclusiveness, COVID-19 vaccines, as common pool resources, present low exclusiveness in the following three aspects: (1) Low exclusiveness of consumption: the open supply of vaccines does not exclude countries without independent production capacity or purchasing power, as all countries are consumers of COVID-19 vaccines. (2) Low exclusiveness of income distribution: it can be predicted that a certain vaccination rate will achieve herd immunity, thus impeding virus spread. A study by the Technion-Israel Institute of Technology has confirmed that vaccination can provide effective protection for unvaccinated people, and for every 20% of the population being vaccinated, the positive rate of rate of unvaccinated people testing positive for COVID-19 will experience a two-fold decline [26]. Vaccination also lowers the infection rates of both vaccinated and unvaccinated people, producing strong positive externalities. (3) Low exclusiveness of decision-making: although countries with independent intellectual property rights over vaccine R&D have stronger decision-making power in vaccine pricing and distribution, other countries still contribute to decisions about supply mode, supply quantity, financing, production, and other vaccine distribution links. As a result, the specific decisions about the supply of vaccines by vaccine-producing countries to external parties under the open supply strategy have in fact shown low exclusiveness.

Judging from the perspective of competitiveness, COVID-19 vaccines, as common pool resources, have manifested extremely high competitiveness. On the one hand, while global open supply has made COVID-19 vaccines accessible to each country in the world, competition between countries still persists, as the yield of vaccines remains limited. This means that the consumption of vaccines by one country will unavoidably reduce the doses of vaccines available for consumption by other countries. On the other hand, within the borders of each country, the consumption of vaccines by some people will unavoidably reduce the doses of vaccines available for consumption by other people.

In summary, under the open supply strategy, COVID-19 vaccines have manifested high competitiveness and low exclusiveness, showing obvious attributes of common pool resources. Thus, as indicated by the correspondence between item specifics and supply modes as shown in Table 2, COVID-19 vaccines should be supplied through global institutional design, i.e., through the coordination and cooperation between countries, as well as between countries and international organizations [22].

## 3. Results of Different COVID-19 Vaccine Supply Strategies

### 3.1. Game Analysis under the Open Supply Strategy

As shown in the left half of Figure 2, the attributes of common pool resources possessed by COVID-19 vaccines under the open supply strategy have turned them into global public goods. This means that countries with independent production capacity are willing to make vaccines equally accessible to other countries or international organizations in a cooperative manner. As shown in Figure 1, in this case, COVID-19 vaccines will have low exclusiveness at the global level (global-wide low exclusiveness, GLE). According to the rules imposed by international organizations (such as COVAX and Global Alliance for Vaccines and Immunisation), vaccines will be distributed according to the population proportions of different countries, so that no vaccine-purchasing country will receive insufficient vaccine doses because of its limited purchasing power. Benefiting from this arrangement, vaccines will manifest low competitiveness among vaccine-purchasing countries (national-wide low competitiveness, NLC).

Moreover, based on the low competitiveness of vaccines in vaccine-producing countries, vaccine-purchasing countries can further distribute the vaccines obtained to their citizens, thus deciding the exclusiveness of COVID-19 vaccines at home. More specifically, vaccine-purchasing countries which have favorable conditions and can purchase or further develop sufficient COVID-19 vaccines can realize the undifferentiated, low-exclusiveness supply of COVID-19 vaccines in their own countries (national-wide low exclusiveness, NLE). In that case, COVID-19 vaccines will turn into low-exclusiveness and low-competitiveness pure public goods in these countries, and their central governments will allocate and supply vaccines to citizens in a unified manner. In contrast, vaccine-purchasing countries that are incapable of purchasing sufficient vaccines to fully satisfy domestic demand because of unfavorable conditions can preferentially supply purchased vaccines to the groups that need them most, thus realizing non-competitiveness of vaccines among these groups. For other groups, COVID-19 vaccines will remain highly exclusive, for a certain period, and will become club goods with national-wide high exclusiveness (NHE) and low competitiveness because of this. Clearly, if COVID-19 vaccines have low exclusiveness at the global level (global-wide low exclusiveness, GLE), they are always accessible to vaccine-purchasing countries. Therefore, when COVID-19 vaccines are high-exclusiveness and low-competitiveness club goods in a vaccine-purchasing country, the vaccine-purchasing country only needs to purchase more vaccines from vaccine-producing countries under suitable conditions and can then supply them as club goods to the next batch of groups in need of vaccination, thus realizing the long-term dissemination of vaccines in its own country.

### 3.2. Game Analysis under a Non-Open Supply Strategy

According to the right half of Figure 2, if COVID-19 vaccines are monopolized as national private goods and supplied according to a non-open supply strategy at the global level, vaccine-producing countries will be reluctant to work with each other on vaccine supply and will implement a nationalistic strategy to monopolize vaccine yield for exclusive use by their own citizens while denying access to other countries [27,28]. As a result, COVID-19 vaccines will manifest high exclusiveness at the global level (global-wide high exclusiveness, GHE). Under the market mechanism, countries will bid for vaccines based on their purchasing power. Vaccine-purchasing countries with weak purchasing power are unlikely to obtain sufficient vaccines, in which case, their citizens will worry about vaccine accessibility and engage in open plunder of limited vaccines, resulting in even higher competitiveness of vaccines (national-wide high competitiveness, NHC). Under such circumstances, no vaccine distribution strategy taken by vaccine-purchasing countries at home will be able to produce desirable results. Concretely, if a vaccine-purchasing country adopts a high-exclusiveness vaccine supply mode at home, COVID-19 vaccines will manifest national-wide high exclusiveness (NHE) and high competitiveness in that country and become the private goods of individuals with a higher payment capacity under the market mechanism. In that case, “a gap between the poor and the rich” will emerge in terms of vaccine distribution, which will ultimately lead to social contradictions. In contrast, if a vaccine-purchasing country adopts a low-exclusiveness vaccine supply mode at home, COVID-19 vaccines will become common pool resources with national-wide low exclusiveness (NLE) and national-wide high competitiveness (NHC) in this country and end up being bought up immediately. In that case, COVID-19 vaccines will be trapped in “the tragedy of the commons” because of undersupply.

### 3.3. Evaluation on Different COVID-19 Vaccine Supply Strategies

Summarizing the game results of COVID-19 vaccines under different supply strategies shows that if COVID-19 vaccines are openly supplied as global public goods, vaccine-purchasing countries will be able to formulate their own anti-pandemic strategies with ease and according to their national conditions. Consequently, they can gradually relieve their citizens from the threat imposed by the pandemic. In contrast, if COVID-19 vaccines are supplied in a non-open manner as national private goods, vaccine-purchasing countries will either be caught up in social contradictions or face extreme scarcity of COVID-19 vaccines. Apparently, COVID-19 vaccines can only substantially contribute to the global fight against COVID-19 when they are openly supplied as global public goods.

## 4. Institutional Guarantee for the Global Supply of COVID-19 Vaccines

When COVID-19 vaccines are treated as low-exclusiveness global public goods, they have essentially become low-exclusiveness and high-competitiveness common pool resources at the global level. The intrinsic susceptibility of common pool resources to the trap of “the tragedy of the commons” means that a global institutional system is needed to guarantee the global supply of COVID-19 vaccines. Thus, global institutional design must be conducted to prevent COVID-19 vaccines with the attributes of common pool resources from falling into the trap of “the tragedy of the commons”. This can guarantee the orderly supply and fair availability of COVID-19 vaccines at the global level [29]. At the level of specific institutional design, in reference to Ostrom’s eight design principles (DP), a global supply institution system of COVID-19 vaccines can be designed in principle [25].

The eight design principles can promote the sustainable management of common pool resources [30], and Table 3 presents the specific principles, which include the following: (1) Clearly defined boundaries: the boundaries of the resource system and the individuals or households with rights to harvest resource units are clearly defined. (2) Proportional equivalence between benefits and costs: rules specifying the amount of resource products that a user is allocated is related to local conditions and to rules requiring labor, materials, and/or money inputs. (3) Collective-choice arrangements: many of the individuals affected by harvesting and protection rules are included in the group who can modify these rules. (4) Monitoring: monitors, who actively audit biophysical conditions and user behavior, are at least partially accountable to the users and/or are the users themselves. (5) Graduated sanctions: users who violate rules-in-use are likely to receive graduated sanctions from other users, from officials accountable to these users, or from both. (6) Conflict-resolution mechanisms: users and their officials have rapid access to low-cost, local arenas to resolve conflict among users or between users and officials. (7) Minimal recognition of rights to organize: the rights of users to devise their own institutions are not challenged by external governmental authorities, and users have long-term tenure rights to the resource. (8) Nested enterprises: appropriation, provision, monitoring, enforcement, conflict resolution, and governance activities are organized in multiple layers of nested enterprises, where smaller-scale organizations tend to be nested in ever larger organizations. The most important implication from the principle of nested enterprises is that institutions can be longitudinally divided into multiple layers, and institutional design can be performed at each layer based on the eight design principles.

Overall, the supply of COVID-19 vaccines exerts its effects at both global and national levels. Therefore, according to DP8 in Table 3, the institutional design that can realize the global supply of COVID-19 vaccines should be a nested rules design conducted at two different levels, namely, the global level and the national level. Figure 3 presents the connotation, structure, and operation mode of the global supply institution. According to DP8 in Table 3, the institutional design that can realize the global supply of COVID-19 vaccines should follow a nested rules design conducted at both global and national levels. Moreover, institutional design at each level is inevitably affected by other external factors such as natural material conditions and socio-economic attributes, the effects of which will be fed back to different levels. Thus, the entire nesting institution for the global supply of COVID-19 vaccines must be constantly adjusted to realize continuous operation.

In the following section, discussions about global institutional design unfold according to DP8 at global and national levels. At the global level, all design principles will be combined to propose institutional suggestions with guiding significance. At the national level, discussions will still be conducted around collective choice rules and operational rules according to DP8. All design principles again will be combined in these two dimensions to develop institutional suggestions with guiding significance.

### 4.1. Principles of Institution Design at the Global Level

#### 4.1.1. DP 1 and 2: Maintaining the Non-Exclusive Supply of COVID-19 Vaccines

As can be concluded from the above sequential game analysis, the best way to advance the global fight against COVID-19 is to treat COVID-19 vaccines as global public goods and supply them non-exclusively at the global level. In this sense, the global institutional design for the supply of COVID-19 vaccines should adhere to the most fundamental principle, i.e., relaxing the accessibility boundary to guarantee the non-excusive supply of COVID-19 vaccines. In the global fight against COVID-19, COVID-19 vaccines are the resources, and mankind is the resource user. In the foreseeable future with insufficient vaccine production capacity, the resource users in various countries are high-risk groups with the right of preferential access, such as medical staff, customs officers, and quarantine workers. Efforts should be made to avoid circumstances where high-income countries seize exclusive monopoly relying on their economic strength and jeopardizing the fair access of low-income countries to vaccines. In this regard, active participation in and management of the COVAX initiative constitute an important path for guaranteeing the global access of COVID-19 vaccines. Related international organizations and vaccine-producing countries also play vital roles in deciding the use boundary and global accessibility of COVID-19 vaccines. In summary, international organizations and countries should be organized at the global level to reach a consensus regarding the global non-exclusive supply of COVID-19 vaccines, and gradually establish the optimal rule of “national population-based distribution”.

#### 4.1.2. DP 3: Realizing Joint Participation of Various Countries in Rule Making

The global non-exclusive supply of COVID-19 vaccines affects their competitiveness in vaccine-purchasing countries. Therefore, vaccine-producing countries, beneficiary countries, and other related stakeholders should be encouraged to jointly participate in the making of the rules governing the global non-exclusive supply of COVID-19 vaccines. Then, the institution designed at the global level can meet the anti-pandemic demands of various countries to the greatest extent. This can be realized through joint participation in the COVAX initiative, i.e., vaccine-producing countries can take part in COVAX to provide surplus vaccines. High-income countries without vaccine production technologies can obtain vaccines through making donations and helping COVAX sign purchase contracts with vaccine producers. COVAX then distributes limited vaccine resources to high-risk groups of countries all over the world based on 20% of their national population. Various countries voluntarily participate in COVAX and independently decide whether to play a role in the global supply institution system regulated by COAVX. They are also free to choose specific manners of participation. In addition, global conferences (such as the World Health Assembly) can also serve as ideal carriers for realizing the institution of collective choice. Global summits and forums can give fuller play to the positive externalities of institutions and make it more convenient for the rules jointly established by various countries to maximally benefit all cooperative partners and other stakeholders, thus realizing rational rule making at the global level [31].

### 4.2. Principles of Institution Design at the Level of National Collective Choice

#### 4.2.1. DP 1 and 2: Relaxing the Technical Boundary between Countries to Improve Vaccine Production Capacity

Basically, the doses of COVID-19 vaccines determine the effectiveness of the global supply institution system. The most effective way to improve vaccine production capacity lies in relaxing the knowledge boundary of vaccine production between countries and promoting international cooperation regarding vaccine production. On the one hand, countries can cooperate in the field of vaccine technologies to accelerate vaccine R&D, reduce R&D costs, facilitate knowledge exchange, and increase the number of varieties of effective vaccines [32,33]. On the other hand, such cooperation also activates related production lines in non-vaccine-producing countries. That is, vaccine-producing countries can export vaccine production technologies to non-vaccine-producing countries, while the latter can leverage their existing resources for vaccine production to improve vaccine production capacity through cooperation. Cooperation between countries in the fields of vaccine production technologies and production lines endows certain vaccine-purchasing countries with the ability to independently produce COVID-19 vaccines, which in turn increases the total vaccine production capacity of the world [34]. The resulting increase in vaccine production capacity further pushes the boundary of resource users in DP1 and promotes the institutional transformation of global vaccine supply towards “national population-based distribution” (DP2).

#### 4.2.2. DP 4, 5 and 6: Promoting the Formation of Supervision, Sanction, and Conflict-Resolving Mechanisms between Countries

In general, the stable operation of an institution requires effective supervision. In the case of the global institution, supervision by various sovereign states and international organizations (such as COVAX) is essential. For instance, the global vaccine supply process should be supervised to see if any country or vaccine producer has exceeded capacity of vaccine in violation of rules. In circumstances where a country intercepts or seizes the vaccines purchased by other countries, signs a purchase agreement with vaccine producers in private, stockpiles vaccines, or engages in other contradictions, countries must negotiate corresponding rules to restrict such contradictions. Where a contradiction cannot be resolved between countries, it should be submitted for arbitration at the global level based on the established contradiction-resolving mechanism. Effective supervision and sanction rules between countries will boost cooperation between countries and improve the smoothness of global vaccine supply. The global reputation of countries can also become the object of supervision and sanction to encourage more countries to play active roles in driving each other’s anti-pandemic efforts through different forms of cooperation.

#### 4.2.3. DP 7: Respecting the Sovereignty and Rights of Each Country and Safeguarding Them against Challenges by External Power Systems

Cooperation between countries and participation of countries in global rule making are both based on a major premise, i.e., each country should be capable of independently designing their own institutions. This means that, in the formation of supply institutions at the national level, the autonomy of various countries as vaccine owners should be respected and protected from external challenges. On the one hand, each country should be free to decide whether to participate in the institutional design between countries. Their autonomy and will of involvement should be safeguarded, and their vaccine distribution at home should not be compromised by external power systems. On the other hand, the national sovereignty of each country should be fully respected, to create a fair and sound mechanism for the cooperation between countries in the global supply system of COVID-19 vaccines.

### 4.3. Principles of Institution Design at the Level of National Operating Rules

When the non-exclusiveness of COVID-19 vaccines at the global level is guaranteed through global institution design and consolidating the competitiveness of COVID-19 vaccines in vaccine-producing countries through institution design between countries, the next step vaccine-purchasing countries face is the selection of a suitable strategy for supplying COVID-19 vaccines to their citizens. To do so, they need to perform institutional design based on local realities.

(1) DP 1 and 2: if the purchasing power of a vaccine-purchasing country is sufficiently strong to support the purchase of sufficient vaccines, the vaccine-producing country should relax its control over vaccine accessibility, i.e., it should adopt a low-exclusiveness strategy for vaccine supply at home. In contrast, if the purchasing power of the vaccine-purchasing country is not sufficiently strong, the country should prioritize vaccine access based on the urgency of demand in different groups. (2) DP3: considering that the vaccine supply of a country relates to the life and health of its citizens, the country should allow full participation from citizens (citizen representatives), thus ensuring that citizens will offer active cooperation and joint participation in vaccine supply and anti-pandemic affairs. (3) DP4, 5 and 6: judiciary organs, discipline inspection and supervision authorities, and community organizations should be urged to fulfill their obligations related to supervision, sanction, and conflict-mediating, thus ensuring the orderly and smooth progress of domestic vaccine supply. (4) DP7: in the anti-pandemic process, the fundamental rights of individuals should be protected from infringement. Countries should encourage their citizens to participate in vaccine use and anti-pandemic efforts through friendly consultations and mutual benefits.

## 5. Conclusions and Discussion

Providing COVID-19 vaccines as global public goods to various countries around the world offers the best choice for boosting the global fight against COVID-19. This approach, however, also must be guaranteed by global institutional design. The fair distribution of COVID-19 vaccines worldwide is crucial for the life and health of all citizens in the world in the short run. In the long run, the fairness of vaccine supply affects the reshaping of the global public health system and the recovery of the global economy. Thus, the global fair distribution of COVID-19 vaccines essentially requires cooperation between countries to realize the supply of high-quality global public goods.

COVID-19 is a public health crisis threatening the common destiny of mankind; therefore, all countries around the world should actively shoulder their responsibilities and work closely with each other in the fight against COVID-19. Notably, to provide COVID-19 vaccines as global public goods, cooperation between countries and their full participation in global institutional design must also be promoted, so that COVID-19 vaccines can ultimately serve the whole world as global public goods.

Guided by Ostrom’s eight design principles, this paper proposes certain principles of institution design for guaranteeing the global supply of COVID-19 vaccines. On the one hand, these principles contribute referential cases to the existing global supply of COVID-19 vaccines. On the other hand, they represent an attempt at applying Ostrom’s eight design principles to problems beyond the scope of traditional common pool resource governance, and an exploration of the possibility of further expanding public affairs governance to the global level.

The proposed multilayer nesting institution for guaranteeing the global supply of COVID-19 vaccines can solve a series of focal problems vaccine supply currently faces.

The first problem relates to the equitable distribution of vaccines between developed and developing countries, which requires coordination between them at the global level. To deal with this problem, first, it must be ensured that COVID-19 vaccines are non-exclusiveness worldwide and fairly accessible by any country. The coordination of vaccine distribution between countries must be directed by an entity acceptable to all parties concerned. In this regard, influential international organizations and responsible world powers should actively shoulder the responsibility. Thus, the cooperation and monitoring between countries can be promoted, the equitable distribution of vaccines worldwide can be facilitated, and nationalism on the issue of vaccine distribution would hopefully be mitigated.

Second, pointed out in proposing the multilayer nesting institution for guaranteeing the global supply of COVID-19 vaccines, this global supply not only requires coordination and monitoring between countries at the global level, but sound institutional rules must also be established by various countries within their respective territories. In this sense, making COVID-19 vaccines accessible in a fair manner to people all around the world is a responsibility for each country, in addition to influential international organizations and responsible world powers. This means that a paramount task for each country in fighting COVID-19 is to conduct political management that suits their national conditions.

The third problem focuses the lack of public confidence in vaccination in some countries. In every country where vaccine distribution is needed, the disclosure of information about vaccine supply and use should be strengthened via political management institutions and platforms. Thus, the effectiveness and safety of vaccines can be presented through changes in vaccination and infection rates to enhance public confidence in vaccination.

Finally, some countries lack the resources to support the popularization of vaccines because of their outdated healthcare systems and poorly developed cold-chain logistics. In this regard, the presented multilayer nesting institution not only promotes the global production and supply of vaccines, but it also supports the cooperation between countries during the transport of vaccines worldwide. Provided that countries all over the world are willing to open technology boundaries and promote international collective action in vaccine supply cooperation, the demands of less developed countries for the cold-chain transport of vaccines can thus be fully satisfied.

## Figures and Tables

**Figure 1 ijerph-19-01339-f001:**
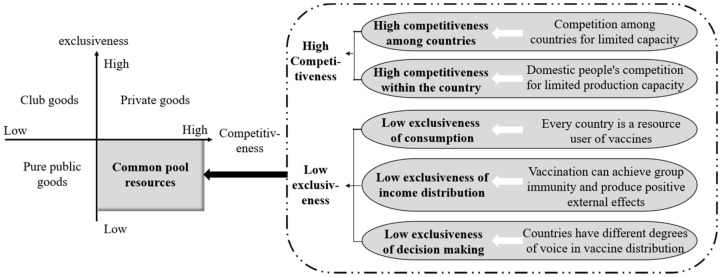
Common pool resources’ attributes of COVID-19 vaccines under open supply strategy.

**Figure 2 ijerph-19-01339-f002:**
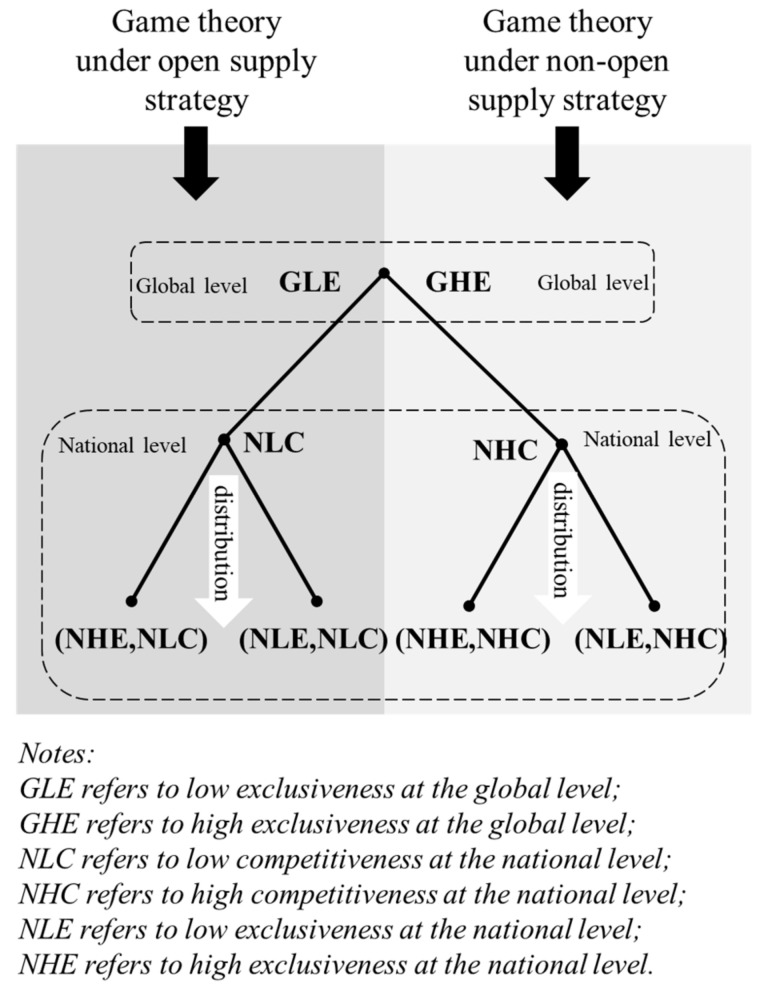
Results of different supply strategies of COVID-19 vaccines.

**Figure 3 ijerph-19-01339-f003:**
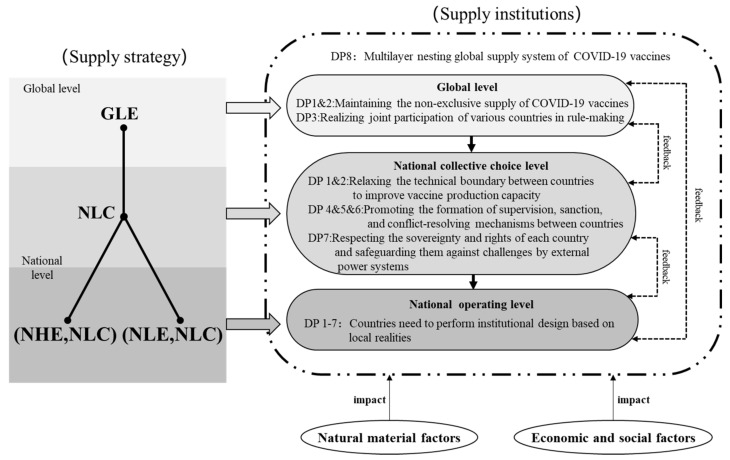
Nested institution to ensure the global supply of COVID-19 vaccine.

**Table 1 ijerph-19-01339-t001:** Classification of economic goods.

		Competitiveness	
		Low	High
exclusiveness	Low	Pure public goods	Common pool resources
	High	Club goods	Private goods

**Table 2 ijerph-19-01339-t002:** Supply mode of economic goods.

Classification of Economic Goods	Representative Figure	Dominant Supply Mode
Pure public goods	Samuelson	Government supply, Joint supply
Club goods	Buchanan	Government supply, Private supply
Common-pool resources	Ostrom	System design according to local conditions
Private goods	Adam Smith	Market supply

**Table 3 ijerph-19-01339-t003:** Ostrom’s eight design principles (adapted from Ostrom 1990).

Number	Design Principles	Content
DP1	Clearly defined boundaries	The boundaries of the resource system and the individuals or households with rights to harvest resource units are clearly defined.
DP2	Proportional equivalence between benefits and costs	Rules specifying the amount of resource products that a user is allocated are related to local conditions and to rules requiring labor, materials, and/or money inputs.
DP3	Collective-choice arrangements	Many of the individuals affected by harvesting and protection rules are included in the group who can modify these rules.
DP4	Monitoring	Monitors, who actively audit biophysical conditions and user behavior, are at least partially accountable to the users and/or are the users themselves.
DP5	Graduated sanctions	Users who violate rules-in-use are likely to receive graduated sanctions from other users, from officials accountable to these users, or from both.
DP6	Conflict-resolution mechanisms	Users and their officials have rapid access to low-cost, local arenas to resolve conflict among users or between users and officials.
DP7	Minimal recognition of rights to organize	The rights of users to devise their own institutions are not challenged by external governmental authorities, and users have long-term tenure rights to the resource.
DP8	Nested enterprises	Appropriation, provision, monitoring, enforcement, conflict resolution, and governance activities are organized in multiple layers of nested enterprises.

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
