# Peer review of "Common Demand vs. Limited Supply—How to Serve the Global Fight against COVID-19 through Proper Supply of COVID-19 Vaccines"

_ijerph, 2022, doi:10.3390/ijerph19031339_

Round 1
Reviewer 1 Report
Dear author/editor,
The manuscript is well written. The argument is valid and time demanded. I recommend publication; however, I suggest incorporating answers to the following questions that I believe would add value to the publication.
- Give your recommendations about the need of a call from the international health care authorities for equal distribution of vaccines in both developed and under-developed countries
- How can the vaccine nationalism can be minimized. Do you have any proposal?
- Political management of COVID-19 would be a problem for vaccination and this might delay the vaccine rollout which will create opportunities for creation of new variants in countries with frail healthcare system. How can global leadership play role in this regard?
- Still there are many countries where vaccine hesitancy is much higher, is there any proposed actions to reduce this problem?
- Some countries do not have sufficient logistics to rollout the rapid vaccination program due to their very frail healthcare systems, thus, there is a chance to produce new variants, how the global leaderships can play their roles to support them?
- These questions need to be solved first to ensure the equal distribution of Covid-19 vaccines across the nations. Have a good luck for this publication!
Reviewer 2 Report
I notice the affiliations are incomplete and/or have errors.
This paper tackles a very interesting question but the paper was not clear to me in many regards. I understand that the paper compares the national private goods model with global public goods and uses game theory to explore vaccination supply strategies.
In the abstract; I’m not sure that the use of the term “low exclusiveness supply” would be clear to all readers. I see that there is no mention of game theory, and also suggest perhaps a little more information about Ostrom’s design principles.
Introduction line 36 “in this sense” doesn’t seem right to me. Perhaps you mean consequently
Line 38; The phrase is “normal order”.
Line 116 should we have some commentary about private goods in this paragraph?
Line 121 please rewrite the phrase “transferred into each other”.
Figure 1 is quite interesting results
Results
line 188 I didn’t follow the reasoning here: please expand.
From line 192 onwards a number of acronyms are introduced which I couldn’t see having been defined earlier. Consequently, I really couldn’t follow this section without understanding what is W DP, CDJ, CDP. I’ve looked back in the paper and can’t see reference to these at all. Hence I was not really able to understand this section.
Line 228 there is a word missing
line 264 perhaps some further description of Ostrom’s 8 design principles or at least some description about nested enterprises
line 275; before 4.1, perhaps it would be worth explaining what the next sections is going to do that is travel through the 8 design principles in the case of covert vaccination distribution.
Line 388 I couldn’t understand the use of the word “attack” in this figure. I would like to see an explanation of figure 3.
Reference 30: is there an author to this paper in the Lancet?
Round 2
Reviewer 2 Report
This manuscript has been greatly improved. I still noted couple of small things. Line 100: ‘private goods are both exclusive and competitive or private goods have the qualities of both exclusiveness and competitiveness….”
Line 129; transferred into each other is still awkward. I think what you mean is economic goods can be transferred into other classifications isn’t it?
Please spell it out table 3
The section from line 456 provides helpful clarity
I think this paper should now be published
